# Use of Ethanol Injections to Create a Complete Atrioventricular Block in a Rat Model

**Abdelmotagaly Elgalad** [1],*, **Ahmed E. Hanafy** [1], **Angel Moctezuma-Ramirez** [1], **Allison Post** [2], **Mathews John** [2], **Yutao Xi** [1] **and Mehdi Razavi** [2]

1 Cardiovascular Surgery Research Lab, Center for Preclinical Surgical and Interventional Research, The Texas Heart Institute, 6770 Bertner Avenue, Houston, TX 77030, USA
2 Electrophysiology Clinical Research and Innovations, The Texas Heart Institute, Houston, TX 77030, USA
* Correspondence: aelgalad@texasheart.org; Tel.: +1-8323557245

**Abstract:** Complete atrioventricular block (AVB) is an abnormal heart rhythm resulting from a defect in the cardiac conduction system. Patients with complete AVB are at risk of symptoms ranging from syncope or hypotension to cardiovascular collapse or sudden cardiac death. A reliable animal model of complete AVB is essential for understanding the mechanisms underlying the fatal hemodynamic effects and alterations in electrical conductivity associated with this arrhythmia. We evaluated the use of ethanol injections in a systematic surgical approach to create a complete AVB model in rats. We used eight Sprague Dawley rats (8 weeks old, 220 ± 30 g): four received a 70% ethanol injection in the AV node, and four received a similar injection of 0.9% sodium chloride. Our surgical approach involved performing a partial sternotomy, using the epicardial fat as a landmark for ethanol injections. Animals were followed for 7 and 14 days. Complete AVB was successfully induced in all four rats that received ethanol injections. Rats in the control group experienced a transient AVB with a return to sinus rhythm. Our study found that using 70% ethanol injections in a systematic surgical approach is a reliable, safe, and reproducible way of creating a complete AVB model in rats.

**Keywords:** complete atrioventricular block; ethanol injection; animal model

## 1. Introduction

Complete atrioventricular block (AVB) is an abnormal heart rhythm caused by a defect in the cardiac conduction system [1]. It occurs when there is no conduction through the atrioventricular node, which results in the atria and ventricles beating independently of each other [2]. Complete AVB can have various causes, including myocardial infarction; drug toxicity; degenerative, infectious, or idiopathic diseases; or cardiac surgery [3]. Patients with complete AVB are at risk of a range of symptoms, such as syncope or hypotension, cardiovascular collapse, or even sudden cardiac death, making it a severe and potentially fatal condition.

Given the high mortality rates associated with complete AVB, developing a reliable animal model to simulate this arrhythmia is crucial for understanding the mechanisms underlying the fatal hemodynamic effects and alterations in electrical conductivity that characterize this condition. An effective animal model of complete AVB can provide a valuable platform for studying the disease's pathogenesis, identifying potential therapeutic targets, and developing new treatments that may improve patient outcomes. Such a model can also aid in the testing of new drugs and medical devices for use in treating this condition. Overall, developing a reliable animal model of complete AVB is essential to improving our understanding of this potentially fatal condition and developing more effective therapies.

Ongoing efforts for developing a complete AVB model [4] are hampered by the need for specialized equipment not available in all laboratories. Several large animal models have been developed. The canine model [5] was established around 1981 and is based on an electrode catheter technique for ablation. In the swine [6] and ovine [7] models, a

radio-frequency ablation approach has been used, which offers a great opportunity for testing tissue engineering and therapeutic efforts. Small animal models for this disease are also common.

In early surgical studies, Boucher et al. injured the AV node by crushing it with forceps after a right atriotomy in a canine model [8]. Other efforts [9] included the use of a thoracotomy without an atriotomy, yielding similar results by electrocauterization of the AV node. Other researchers introduced a new method that did not require a surgical approach by using a catheter to inject formalin directly into the vicinity of the AV node [10].

Models in rodents, including transgenic breeds [11,12], have used radiofrequency ablation of the AV node, along with nodal injections of 70% alcohol [13]. At our center, we have used pacemaker stem cells, carbon nanotube fiber conductivity [14], and conductivity changes in hydrogels for myocardial pacing and restoring myocardial conductivity.

In the past, ethanol injection [15] was a common method used to induce AVB in animal models but is no longer considered a viable option because of its limitations. Investigators have only a limited number of injection attempts, as multiple injections may lead to permanent damage to a broad area of the heart, which can result in global cardiac dysfunction. Thus, other methods for inducing AVB have emerged. Animal models have become essential in understanding the disease's hemodynamics, physiological adaptations, and electrophysiology. These models provide a platform in which therapies can be tested, including gene therapy, cell therapy, biological pacemakers, or novel materials for cardiac resynchronization therapy. By using animal models, researchers can gain a better understanding of the mechanisms underlying AVB and the potential treatments that may be effective. Additionally, animal models allow for the testing of new treatments that may not be safe or feasible to test in humans. Therefore, animal models play a vital role in the study and treatment of AVB, providing valuable insights into the mechanisms underlying this disease and offering hope for new treatments that may improve patient outcomes.

In this study, we found that a systematic surgical approach using ethanol injections is a reliable, safe, and reproducible way to induce complete AVB in a rat model.

## 2. Materials and Methods

All surgical and animal care protocols used in this study were approved on 29 March 2018, by the institutional animal care and use committee at The Texas Heart Institute. A total of 8 Sprague Dawley rats (8 weeks old; average weight, $220 \pm 30$ g) were included in this study: 4 rats received a 70% ethanol injection in the AV node, and 4 rats received a similar injection of 0.9% sodium chloride (controls). Animals were followed for 7 and 14 days; half of the rats in each treatment group were allocated for each time period. In preparation for this protocol, 3 animals were allocated for training in the surgical technique.

For the surgical approach, a sterile field of supplies was prepared, and the rats were shaved and prepared. The rats were anesthetized in an induction Pyrex chamber with 5% isoflurane in 100% oxygen at a constant flow rate of 2 L/min for 5 min. Endotracheal intubation was performed with a 16-gauge, 50.8 mm long intravenous catheter. Ventilation was administered through a respirator (VentElite Small Animal Ventilator, Harvard Apparatus, Holliston, MA, USA) at 80 breaths per minute with a 2 mL tidal volume. Anesthesia was maintained with 2% isoflurane mixed with 100% oxygen for the duration of the procedure. Bipolar, limb-lead surface electrocardiograms were recorded on the anesthetized animals with external electrodes (Standard 1.5 mm DIN ECG leads, INDUS Instruments, Webster, TX, USA). A partial sternotomy, conserving the manubrium, was performed to expose the heart. The area of the xiphoid process was dissected to create a fine hole in between the xiphoid edges to allow for the insertion of one blade of the scissors to cut the lower part of the sternum. The cutting was continued until the manubrium was reached. A bovie pen (battery-operated electrocautery) can be used to minimize bleeding from the sternal edges and surrounding muscles. The pericardium was incised, and a cotton tip swab was swiped laterally. The thymus gland and pericardial pad of fat were also swiped with a cotton swap; avoiding cutting the fat and thymic tissue helps to prevent unnecessary bleeding.

The right atrial appendage was retracted by using fine forceps or peanut-shaped gauze (which is better) to expose the groove along the medial wall of the right atrium and the root of the ascending aorta (Figure 1). A small fat pad was localized and served as an anatomical landmark. A syringe with a 27 G needle with a 90° bend located 5 mm from the tip was filled with 5 µL of 70% ethanol and inserted parallel to the aortic wall axis and perpendicular to the myocardial tissue just beneath the fat pad. The needle was turned 30° toward the tricuspid valve until a 5 mm depth was achieved. An electrocardiogram was continuously obtained using a rodent surgical monitor (Indus Instruments).

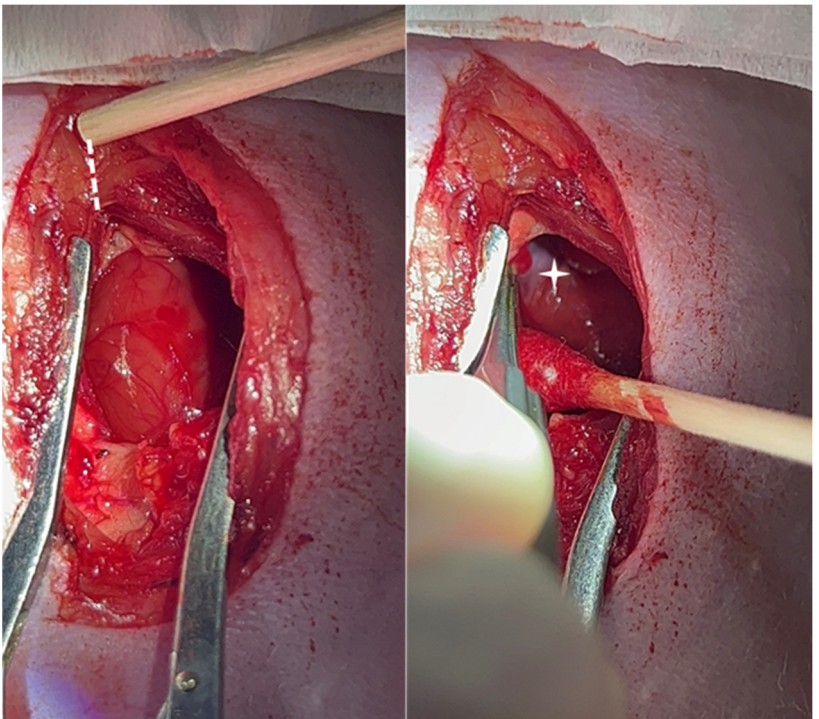

**Figure 1.** Partial sternotomy with preservation of the manubrium (right panel, white dotted arrow). The right atrial appendage is retracted to expose the groove along the right atrium and the root of the ascending aorta (left panel, asterisk).

When a prolonged PR interval was noted on the electrocardiogram, the solution was injected to ablate the AV node. Up to 3 injections were used to confirm complete AVB; the total injected volume did not exceed 15 µL. Control rats received injections of 5 µL of 0.9% sodium chloride under the same surgical protocol.

Once complete AVB was achieved continuously for at least 5 min, we allowed for acclimation of the new heart rhythm and cardiac output before closing the thoracic cavity. A 16 G × 50.8 mm catheter was inserted into the open chest cavity to serve as a chest tube, and a hemostat was used to secure it. The chest cage edges were brought together with a 4-0 chromic gut suture, and the muscle layer was closed with 7-0 Vicryl plus suture.

Lastly, 5-0 Prolene was used to suture the skin. A 3 mL syringe was connected to the catheter in the chest cavity, the hemostat was unclamped, and suction was created by pulling on the plunger while slowly removing the syringe with the catheter from the chest cavity.

Post-surgical analgesia consisted of buprenorphine (1 mg/kg), given subcutaneously. The animals were observed until the day of euthanasia, and no complications were noted during the follow-up period. The rats were euthanized at 7 and 14 days after surgery (groups 7 and 14 days, respectively). The procedure involved the same surgical approach through median sternotomy under deep anesthesia. Potassium chloride was injected directly into the inferior vena cava to stop the heart during diastole. Euthanasia was

confirmed by observing continuous asystole on the electrocardiogram monitor. Before euthanasia, we confirmed either AVB or sinus rhythm on an electrocardiogram. The heart was removed for pathologic study.

The hearts were excised and fixed in 10% neutral buffered formalin for 24–48 h. After fixation, the two lower thirds of the ventricles were discarded. Using a dissecting microscope, we trimmed the remaining base of the heart. The base was then processed by dehydration through graded alcohols and xylene and embedded in paraffin. We cut 5 μm thick tissue sections perpendicular to the tricuspid valve annulus at 200 μm intervals and stained them with hematoxylin and eosin. Immediately adjacent sections were stained with Masson's trichrome.

## 3. Results

A stable complete AVB was induced in all four ethanol-injected rats; the heart rate was $375 \pm 55$ beats per minute (bpm) before ethanol injections and $120 \pm 40$ bpm after injections. No major complications were observed during the procedure or the postoperative period (100% survival rate). No sternal dehiscence, respiratory issues, or wound complications were noted during the follow-up period. AVB in the study group was confirmed by surface electrocardiogram. Once the needle was inserted toward the AV node, supraventricular tachycardia ensued (Figure 2). After the ethanol injections, a complete AVB was achieved (Figure 3). In control rats, repeated saline injections showed that mechanical poking of the AV node was insufficient to create a stable complete AVB; sinus rhythm usually returned within 2 to 3 min after injection. Figure 4 shows a control rat's transition from incomplete AVB to sinus rhythm.

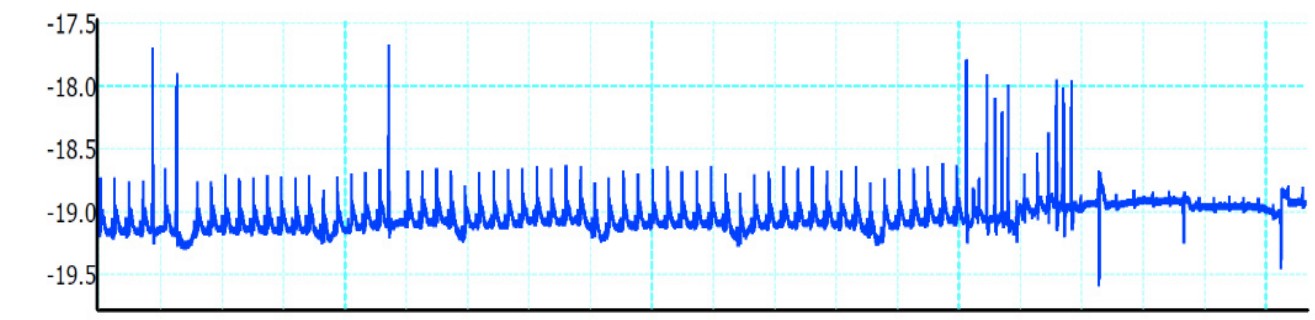

**Figure 2.** Electrocardiogram shows supraventricular tachycardia development as the needle is inserted toward the AV node.

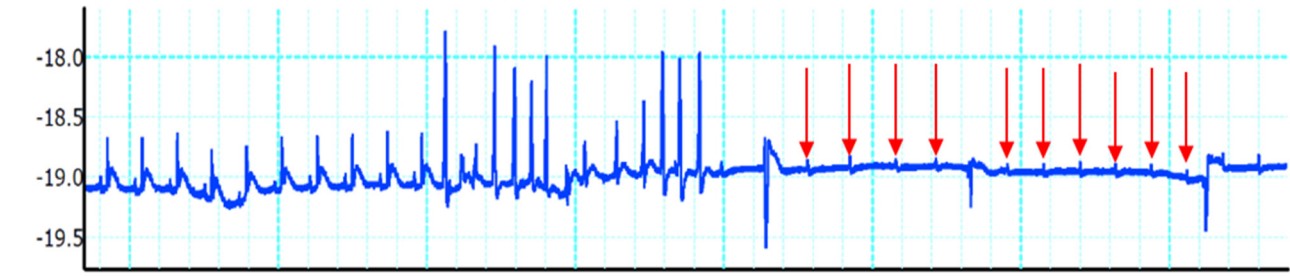

**Figure 3.** Ethanol injection completed; complete AVB is achieved. Arrows indicate blocked p waves.

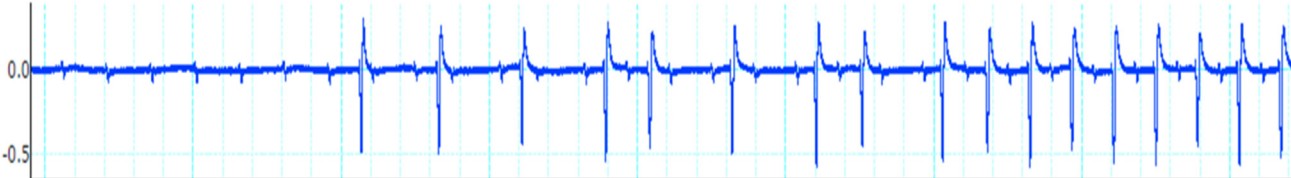

**Figure 4.** In control rats, the saline injection did not create a complete AVB. Incomplete AVB returns to sinus rhythm.

Figure 5 shows a coronal section of an ethanol-injected rat heart stained with Masson's trichrome; the needle trajectory from the epicardial adipose pad to the AV node shows evidence of acute injury as suggested by areas of hemorrhage.

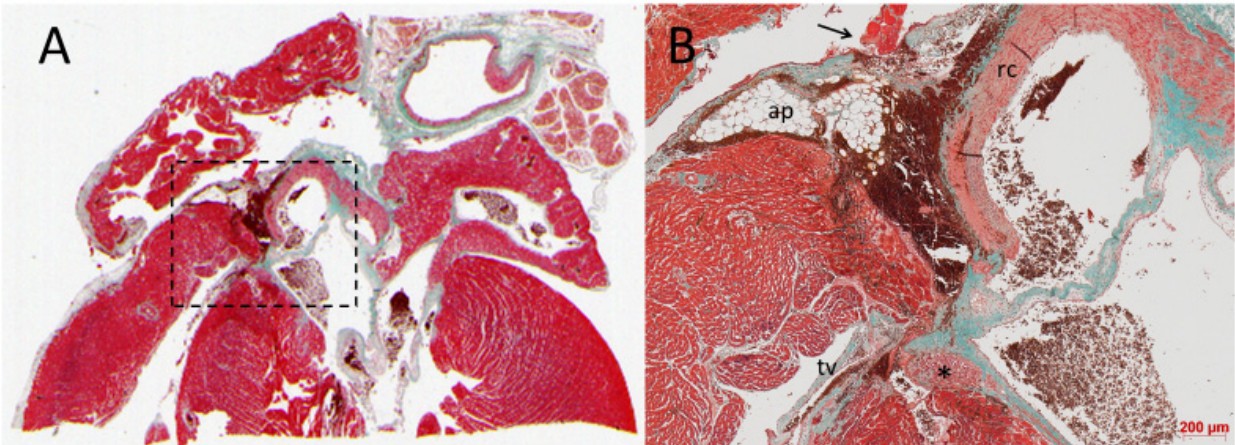

**Figure 5.** (**A**) Panoramic view of the base of the heart. (**B**) Higher magnification of the inset in A. The epicardial adipose pad used as a landmark for the ethanol injections shows a needle entrance wound site plugged by an acute thrombus (arrow). Acute hemorrhage infiltrates much of the area and extends to the surroundings of the AV node (asterisk), which shows acute injury.

## 4. Discussion

Animal models have played a crucial role in providing data that are important for determining the pathogenesis of cardiac arrhythmias. Although various animal models have been used, dogs, pigs, guinea pigs, rabbits, rats, and mice are the most common models. However, ethical considerations and the additional costs of working with larger animals have led to rodents being the preferred choice for laboratory settings. Rodents offer several advantages, such as their availability in many strains worldwide. Using these strains ensures genetic similarity, reduces variability among individual animals, and makes results more easily reproducible. A complete AVB model in rodents is particularly advantageous, as it can provide insights into the pathogenesis of this condition. Additionally, the use of rodents in laboratory settings enables researchers to conduct experiments that would not be possible in humans due to ethical considerations. Therefore, although other animal models are still used, rodents are a crucial part of the study of cardiac arrhythmias.

Genetically modified rodents are also used in the study of arrhythmias. This approach allows researchers to create a specific disease model by inducing genetic modifications that facilitate the appearance of the desired condition. For example, Kim et al. [4] induced complete AVB in rats by using electrosurgical needle ablation and focal somatic gene transfer of TBX18 to the left ventricular apex. Within days, the rats developed ectopic ventricular beats and achieved a de novo ventricular rate faster than the atrioventricular junctional escape rhythm observed in control animals. While this approach has several advantages, including the ability to generate a specific disease model quickly, our model did not incorporate genetically modified rodents. Instead, we aimed to provide a simple

alternative to laboratories seeking to induce complete AVB by using commonly available laboratory supplies. Our approach can be useful for laboratories that do not have access to genetically modified rodents or prefer not to use them for ethical reasons. Overall, the use of genetically modified rodents and other techniques such as electrosurgical needle ablation can provide valuable insights into the pathogenesis of arrhythmias and facilitate the development of effective treatments.

Achieving complete AVB in a preclinical setting can be accomplished using various methods that generally fall into three broad categories: catheter-based approaches, heat delivery, or injection of cytotoxic substances. Catheter-based methods involve using custom-made radiofrequency ablation catheters that are inserted through the right internal jugular vein to reach the right atrium [16]. In the heat delivery approach, electrocautery is applied directly to the AV node area [17]. Injection procedures also use a direct approach into the AV node, with cytotoxic chemicals such as formalin used to achieve ablation [18]. Despite their differences, all these methods rely on the same anatomical landmark—the epicardial fat pad—also used in this study. By using this landmark, researchers can accurately locate and ablate the AV node, resulting in complete AVB. Although these techniques have been shown to be effective in inducing a complete block, they each have their limitations and potential drawbacks. For example, catheter-based approaches can be time-consuming and require skilled operators, whereas heat delivery and injection procedures may cause damage to adjacent structures. Therefore, researchers must carefully consider the advantages and disadvantages of each approach and select the most appropriate one for their specific research needs.

An advantage of our method over techniques that use ablation or radiofrequency is that the electrocardiogram tracing was not affected by interference when the ethanol was delivered and did not require repositioning or further data filtration. This ensures the consistent acquisition of electrocardiographic data. In addition to this, the partial sternotomy in this study further facilitated closure of the thorax.

Ethanol injections have been recently abandoned because of technical challenges and complications. These include bleeding associated with a complete sternotomy and the level of expertise needed for adequate exposure of the base of the heart and identification of the correct anatomical landmarks. Additionally, multiple injections can lead to global myocardial damage, activation of the coagulation cascade in the heart, and even death [4]. Thus, success has been limited with this approach. However, as we have shown in this study, a systematic surgical approach with a partial sternotomy using the epicardial fat as a landmark for ethanol injections can achieve the same results as other methods and avoids the need for custom-made materials, a complex setup, or specific rat breeds. Furthermore, using a partial sternotomy that conserves the manubrium avoids the bleeding complications and high mortality seen in previous surgical models [13,17], as indicated by our 100% survival rate.

The null mortality rate observed in this study can be attributed to the use of a systematic approach in the surgical method. Reproducibility is key when sharing an experimental approach, and we planned ahead for each step of the procedure to ensure clarity. Since techniques such as ablation or ethanol use tend to be limited by the number of ablations or injections, every attempt has to be precise, which we have shown is possible when using this approach. Although closed chest models are also available, they often require the use of specialized catheters that require techniques such as electrical ablation, laser treatment, radiofrequency ablation, or cryoablation.

A limitation of this study is the small number of animals used; a larger experimental population is needed for more robust results and to address individual variability. Studies are required to evaluate more time points, of both shorter and longer duration, which could potentially assess the permanency of the complete AVB.

## 5. Conclusions

We have shown that using the epicardial fat pad between the aortic root and the right atrial wall of a rat's heart as a landmark for 70% ethanol injections is a reliable, safe, and reproducible way for inducing complete AVB in rats. This surgical technique is a valuable tool for creating a model of complete AVB. Furthermore, this model can be validated in electrophysiology and regenerative medicine preclinical research.

**Author Contributions:** A.E.: Conceptualization, investigation, writing—review and editing. A.E.H.: Methodology, investigation, writing—original draft. A.M.-R.: Data curation, writing—original draft, investigation. A.P.: Resources. M.J.: Resources. Y.X.: Writing—review and editing. M.R.: Supervision. All authors have read and agreed to the published version of the manuscript.

**Funding:** This research received no external funding.

**Institutional Review Board Statement:** The animal study protocol was approved by the Institutional Animal Care and Use Committee of The Texas Heart Institute (protocol #2018-05 approved on March 2018).

**Informed Consent Statement:** Not applicable.

**Data Availability Statement:** The data that support the findings of this study are available from the corresponding author, A.E., upon reasonable request.

**Acknowledgments:** The authors wish to thank Rebecca Bartow of Scientific Publications at The Texas Heart Institute for her editorial contributions.

**Conflicts of Interest:** The authors declare no conflict of interest.

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
