# Peer review of "Use of Ethanol Injections to Create a Complete Atrioventricular Block in a Rat Model"

_2673-4095, doi:10.3390/surgeries4020020_

Round 1
Reviewer 1 Report
Re: Surgeries-2338474-peer-review
The brief report (the theme) is not a very new one, however, it has some scientific value, and the manuscript is well written, easy to follow.
The authors studied the ethanol-induced atrioventricular block (AVB), an abnormal heart rhythm, in the cardiac conduction system. The complete AVB is a high risk for the development of arrhythmias, leading to sudden cardiac death. Sprague Dawley rats were subjected to 70% of ethanol injection into the AV node, and the control animals received saline injection (normal). Complete AVB was successfully induced in all rats received ethanol injection. The results show that the use of 70% ethanol injection, under the surgical procedure, is a reliable method for creating complete AVB in rats.
Comments:
In the Abstract: What does it mean that “normal saline” injection?
What is the survival rate? All rats can survive the surgical procedure for several day or weeks. In other words, how many animals had to be used to get e.g., four survivors in each group?
Figure 1. Should be complemented with additional pictures about the different steppes of the surgical procedure for easier understanding and replicability.
What can be the clinical relevance of the ethanol-induced AVB? For instance, in alcoholic patients or alcohol intoxication?
Indeed, many diseases, including myocardial ischemia, myocardial infarction, drug toxicity, idiopathic diseases, ischemia/reperfusion, and cardiac surgery can lead to changes in the action potential duration and electrocardiogram (ECG), resulting in cardiovascular collapse, serious arrhythmias such ventricular fibrillation. The final outcome is the sudden cardiac death. Several factors and mediators cause abnormalities in the ECG, therefore, various interventions are necessary to be used to prevent the sudden cardiac death, preventing the development of e.g., AV block, long QT syndromes ‘torsade des pointes’ arrhythmias, and Brugada syndromes.
Mechanisms of arrhythmias and sudden cardiac deaths have been recently published in valuable journals, which might be briefly acknowledged and cited in a short paragraph in the revised version of this (Surgeries-2338474) manuscript.
These are the followings:
- Eur Heart J. 2023 Mar 21;44(12):1058-1066. doi: 10.1093/eurheartj/ehac799
Eur Heart J Cardiovasc Pharmacother. 2023 Feb 2;9(2):173-182. doi: 10.1093/ehjcvp/pvac069
- JACC Clin Electrophysiol. 2023 Feb;9(2):266-279. doi: 10.1016/j.jacep.2022.10.023
- Physiol Rev. 2021 Jul 1;101(3):1083-1176. doi: 10.1152/physrev.00024.2019. Epub 2020 Oct 29
- PLoS One. 2020 Aug 27;15(8):e0237854. doi: 10.1371/journal.pone.0237854. eCollection 2020
- Front Pharmacol. 2020 May 12;11:616. doi: 10.3389/fphar.2020.00616. eCollection 2020
- Int J Mol Sci. 2019 Apr 29;20(9):2123. doi: 10.3390/ijms20092123
- Front Physiol. 2019 Jan 7;9:1847. doi: 10.3389/fphys.2018.01847. eCollection 2018
- Cardiol J. 2018;25(6):709-713. doi: 10.5603/CJ.a2017.0156. Epub 2018 Jan 3
In summary, the incorporation of the aforementioned publications in the revised version may substantially increase the interest of general readers, the clinicians and experimental researchers as well.
Author Response
Reviewer 1
The brief report (the theme) is not a very new one, however, it has some scientific value, and the manuscript is well written, easy to follow.
The authors studied the ethanol-induced atrioventricular block (AVB), an abnormal heart rhythm, in the cardiac conduction system. The complete AVB is a high risk for the development of arrhythmias, leading to sudden cardiac death. Sprague Dawley rats were subjected to 70% of ethanol injection into the AV node, and the control animals received saline injection (normal). Complete AVB was successfully induced in all rats received ethanol injection. The results show that the use of 70% ethanol injection, under the surgical procedure, is a reliable method for creating complete AVB in rats.
In the Abstract: What does it mean that “normal saline” injection?
Response: The use of the term “normal saline” refers to the use of 0.9% sodium chloride solution. As per the reviewer’s comment, we have now updated the manuscript with this information (see the Abstract and pages 2 and 3 in the Materials and Methods section).
What is the survival rate? All rats can survive the surgical procedure for several day or weeks. In other words, how many animals had to be used to get e.g., four survivors in each group?
Response: The reviewer brings up a good point. Before starting this research protocol, we trained for performing these experiments in 3 animals for surgical practice. After this training, we attained a 100% survival rate. We have now included this point in the manuscript (see the first paragraph of the Materials and Methods section).
Figure 1. Should be complemented with additional pictures about the different steppes of the surgical procedure for easier understanding and replicability.
Response: We appreciate the reviewer’s comment and feedback. Since sternotomies are well documented in small animal models, we did not include images showing the surgical approach. We included photos of the most critical steps of the protocol for the readers: the manubrium and xiphoid process sparing and the identification of the epicardial fat pad.
What can be the clinical relevance of the ethanol-induced AVB? For instance, in alcoholic patients or alcohol intoxication?
Response: As the reviewer infers, the clinical correlation between alcohol and arrhythmias is extensively documented, and its clinical value is of high importance. However, in this context, we did not intend to compare alcohol-induced arrhythmias and the reproducibility of this disease model. The ethanol was used in this model to induce injury to the atrioventricular node and hence atrioventricular block.
Mechanisms of arrhythmias and sudden cardiac deaths have been recently published in valuable journals, which might be briefly acknowledged and cited in a short paragraph in the revised version of this (Surgeries-2338474) manuscript.
These are the followings:
- Eur Heart J. 2023 Mar 21;44(12):1058-1066. doi: 10.1093/eurheartj/ehac799
Eur Heart J Cardiovasc Pharmacother. 2023 Feb 2;9(2):173-182. doi: 10.1093/ehjcvp/pvac069
- JACC Clin Electrophysiol. 2023 Feb;9(2):266-279. doi: 10.1016/j.jacep.2022.10.023
- Physiol Rev. 2021 Jul 1;101(3):1083-1176. doi: 10.1152/physrev.00024.2019. Epub 2020 Oct 29
- PLoS One. 2020 Aug 27;15(8):e0237854. doi: 10.1371/journal.pone.0237854. eCollection 2020
- Front Pharmacol. 2020 May 12;11:616. doi: 10.3389/fphar.2020.00616. eCollection 2020
- Int J Mol Sci. 2019 Apr 29;20(9):2123. doi: 10.3390/ijms20092123
- Front Physiol. 2019 Jan 7;9:1847. doi: 10.3389/fphys.2018.01847. eCollection 2018
- Cardiol J. 2018;25(6):709-713. doi: 10.5603/CJ.a2017.0156. Epub 2018 Jan 3
In summary, the incorporation of the aforementioned publications in the revised version may substantially increase the interest of general readers, the clinicians and experimental researchers as well.
Response: We appreciate the reviewer’s input and have carefully reviewed the reference list. Although we agree in the value of these references and their publication in journals of excellence, we do not believe they align closely enough with the aims of this brief report to be included in the citations. The different etiologies of the mechanisms of arrhythmias are not associated with this disease model since our model involves a mechanical and cytotoxic injury to the AV node.
Reviewer 2 Report
The authors present an interesting study showing the methods for developing a model of complete AV block in rats. I have only a few minor comments. First, do you think this method could be scaled up to larger animal models such as swine with similar success? And second, what do you think the ultimate success rate of this procedure would be with n >> 4? I congratulate you on your success but with only an n=4, it is hard to tell how truly repeatable this method is.
Author Response
Reviewer 2
The authors present an interesting study showing the methods for developing a model of complete AV block in rats. I have only a few minor comments.
First, do you think this method could be scaled up to larger animal models such as swine with similar success?
Response: We thank the reviewer for the positive comments and the thoughtful question. Although this method (or a similar one) has been previously used in large animal models, it is not the best option. There are better minimally invasive methods that do not require opening the chest, such as using commercially available ablation catheters to ablate the atrioventricular node. Custom-made catheters for rodents are expensive and hard to find; pharmacological agents used in rodents require micro-dosing and expert titration.
And second, what do you think the ultimate success rate of this procedure would be with n >> 4? I congratulate you on your success but with only an n=4, it is hard to tell how truly repeatable this method is.
Response: We agree with the reviewer that this is a small sample and a limitation of our study. We do not know the answer to the reviewer’s insightful question since we have not done this protocol in a larger sample size. Thus, more studies with a larger number of animals are needed to address the reproducibility and statistical significance of this approach. To better address this issue, we have now added a limitations paragraph to the Discussion (see page 7).